# Italian Precision Medicine in Pediatric Oncology: Moving beyond Actionable Alterations

**DOI:** 10.3390/ijms231911236

**Published:** 2022-09-23

**Authors:** Fabio Pastorino, Mario Capasso, Chiara Brignole, Serena Giglio, Veronica Bensa, Sueva Cantalupo, Vito Alessandro Lasorsa, Annalisa Tondo, Rossella Mura, Angela Rita Sementa, Alberto Garaventa, Mirco Ponzoni, Loredana Amoroso

**Affiliations:** 1Laboratorio di Terapie Sperimentali in Oncologia, IRCCS Istituto Giannina Gaslini, 16147 Genova, Italy; 2Dipartimento di Medicina Molecolare e Biotecnologie Mediche, Università degli Studi di Napoli Federico II, 80131 Napoli, Italy; 3CEINGE Biotecnologie Avanzate, 80131 Napoli, Italy; 4UOC Oncologia, IRCCS Istituto Giannina Gaslini, 16147 Genova, Italy; 5Dipartimento di Oncoematologia, Ospedale Meyer, 50139 Firenze, Italy; 6Oncoematologia Pediatrica, Ospedale Pediatrico Microcitemico “Antonio Cao” Azienda Ospedaliera Brotzu, 09121 Cagliari, Italy; 7Dipartimento di Patologia, IRCCS Istituto Giannina Gaslini, 16147 Genova, Italy

**Keywords:** whole exome sequencing, precision medicine, pediatric oncology, neuroblastoma

## Abstract

Neuroblastoma (NB) is the most common extracranial solid tumor encountered in childhood. Although there has been significant improvement in the outcomes of patients with high-risk disease, the prognosis for patients with metastatic relapse or refractory disease is poor. Hence, the clinical integration of genome sequencing into standard clinical practice is necessary in order to develop personalized therapy for children with relapsed or refractory disease. The PeRsonalizEdMEdicine (PREME) project focuses on the design of innovative therapeutic strategies for patients suffering from relapsed NB. We performed whole exome sequencing (WES) of patient-matched tumor-normal samples to identify genetic variants amenable to precision medicine. Specifically, two patients were studied (First case: a three-year-old male with early relapsed NB; Second case: a 20-year-old male who relapsed 10 years after the first diagnosis of NB). Results were reviewed by a multi-disciplinary molecular tumor board (MTB) and clinical reports were issued to the ordering physician. WES revealed the mutation c.G320C in the CUL4A gene in case 1 and the mutation c.A484G in the PSMC2 gene in case 2. Both patients were treated according to these actionable alterations, with promising results. The effective treatment of NB is one of the main challenges in pediatric oncology. In the era of precision medicine, the need to design new therapeutic strategies for NB is fundamental. Our results demonstrate the feasibility of incorporating clinical WES into pediatric oncology practice.

## 1. Introduction

Neuroblastoma (NB) is the most frequent extracranial solid tumor in children (10% of cancers), with an estimated incidence of 1/8000–10,000 [1,2]. Although important progress has been made in the field of NB biology, and new drugs and therapeutic modalities have been introduced for the treatment of this tumor, the percentage of patients who experience disease progression, recurrence or drug-resistance remains significantly higher than that observed in other pediatric cancers [3]. High-risk NB (HR-NB) is still the leading cause of cancer mortality in patients under five years of age, being responsible for 15% of all deaths due to childhood cancer [2,3].

Much of the progress in the treatment of NB is the result of the implementation of risk-stratified treatment strategies that incorporate histology, stage, molecular prognostic factors and age on diagnosis into the choice of therapies. Ploidy, chromosomal segmental changes and specific gene alterations are routinely utilized to guide treatment intensity in NB. Therefore, the basic tenets of precision medicine are intrinsic to the practice of pediatric oncology. Recent high-throughput genomic analyses of pediatric tumors have suggested that the identification of somatic and hereditary genetic variants could help to improve diagnosis and prognosis, especially in the case of highly aggressive tumors [4,5,6] and indicated that recurrent somatic mutations are rare in primary NB [4,7,8,9].

By contrast, relapsed and metastatic tumors are rich in actionable mutations, which are potential therapeutic targets [10].

In addition, genetic association studies have shown that rare and common germline genetic variants are associated with metastatic NB [4,7,9,11]. Overall, these studies suggest that a greater understanding of the genomic alterations of metastatic and relapsed forms of NB could have an impact on the patient’s prognosis and response to therapy. Here, tumor biopsies derived from two patients suffering from relapsed HR-NB were processed for DNA characterization by means of next-generation sequencing (NGS).

Recent advances in genetic sequencing in pediatric patients have given rise to more comprehensive approaches to determining the abnormalities that contribute to tumorigenesis [9]. Initial implementation of whole exome sequencing (WES) technologies focused on identifying actionable alterations, which were estimated to be present in 5% to nearly 10% of cases [12]. PREME (PeRsonalizEdMEdicine) is an Italian multi-center, prospective, non-profit project composed of sub-projects that focus on the design of innovative and personalized/precision therapeutic strategies for patients with NB. Here, we report our experience of integrating clinical WES into pediatric oncology practice.

## 2. Results

### 2.1. Case 1

In 2013, a three-year-old male child was referred to our institution following the diagnosis of stage 4 NB by means of biopsy of the right adrenal gland. Physical examination showed no enlarged lymph nodes, the blood cell count was normal, and LDH was mildly increased (249 IU/l, normal value 135-225 IU/l). A bone marrow biopsy revealed disease infiltration, and a computed tomography (CT) scan showed an adrenal mass associated to multiple enlarged celiac lymph nodes. Therapy was administered according to SIOPEN guidelines on HR-NB. The patient underwent induction chemotherapy with COJEC (cyclophosphamide, vincristine, cisplatin, carboplatin, etoposide), without dose delays or dose reductions, high-dose chemotherapy with busulfan and melphalan and autologous hematopoietic stem cell transplantation (HSCT), radical surgery of the adrenal mass, radiotherapy and cis retinoic acid maintenance therapy.

During maintenance therapy, the child relapsed and was treated with chemotherapy, ^131^I-MIBG radiometabolic therapy with tandem high-dose chemotherapy, and immunotherapy with anti-GD2 monoclonal antibodies. Four months after the end of immunotherapy, a central nervous system (CNS) relapse was diagnosed. The patient was again treated with chemotherapy and cerebral radiotherapy (30 Gy). Five months after the end of treatment, he came to the emergency department because of a headache. A brain CT scan showed CNS progression with a right frontal mass that extended to the ipsilateral frontal-basal areas, and a large ipsilateral vasogenic edema with ventricular compression (Figure 1).

Neurosurgical excision was performed, and a piece of the tumor was stored for molecular analysis. WES analysis showed four candidate genes as potential therapeutic targets (Table 1). However, for the reasons explained below, we focused on the mutation c.G320C in the *CUL4A* gene.

Indeed, *CUL4A* over-expression is a common feature of many human cancers, including pediatric medulloblastoma [13]. CNS relapse is an infrequent complication in NB patients. However, when present, it is severe and is associated with a poor prognosis. Moreover, it has been shown that high levels of *CUL4A* expression render prostate cancer cells sensitive to thalidomide [13,14].

On the basis of these premises, two drugs came to our attention owing to their clinical use in CNS tumors: lenalidomide and temozolomide.

Lenalidomide is a promising agent that inhibits angiogenesis by blocking VEGF-induced PI3K-AKT pathway signaling; it also inhibits IL-6 and endothelial cell function. Additionally, lenalidomide can induce tumor cell apoptosis and act as an immune-modulator by stimulating T-cell proliferation, reducing the expression of immune-checkpoint inhibitors on both T and NK cells, inhibiting TNF-alpha and reducing T-regulatory cells. Some interesting reports have shown its efficacy in treating CNS relapse and its ability to cross the blood-brain barrier (BBB) [15,16,17]. Temozolomide, on the other hand, is an alkylating chemotherapy agent that is used in a variety of solid and CNS tumors, and which has excellent BBB penetration [15].

Within the PREME project, the molecular tumor board (MTB) decided to administer lenalidomide at a dose of 25 mg daily for 21 days every 28 days, together with temozolomide at a dose of 250 mg daily for five days every 28 days. The patient was in good clinical condition until the end of the third course, when headache recurred. A gadolinium-enhanced brain MRI showed disease progression. The clinical condition rapidly worsened and the patient died.

It is important to note that, in our case, the combination of temozolomide and lenalidomide was well tolerated in terms of hematologic toxicity.

### 2.2. Case 2

This patient suffered a late recurrence of NB, 10 years after the diagnosis of HR-NB, which had undergone first-line treatment according to the NB-HR-01 protocol, without the use of maintenance anti-GD2 immunotherapy. In September 2009, the five-year-old patient presented with left paravertebral undifferentiated NB, *MycN* not amplified, with multiple skeletal metastases (limbs, right hemipelvis), thoracic lymph node metastases and bone marrow infiltration. We started treatment according to the SIOPEN NB-HR-01 protocol and, after surgery on the primary lesion, he received four courses of TVD, obtaining a partial response (PR). Consolidation therapy with busulfan-melphalan and autologous stem cell rescue was then performed. Radiotherapy was carried out on the primary lesion (21 Gy) followed by maintenance therapy with cis-retinoic acid. The patient had achieved complete remission by the end of therapy and started the follow-up period in March 2011. At the end of April 2020, the patient presented with pain and swelling of the left arm. MRI with minimum detectable change (mdc) revealed a mass that involved the distal humerus and infiltrated the medullary canal for about 9 cm and the soft tissues for about 6 cm. A CT-guided needle biopsy was performed on the suspicion of NB recurrence. Histopathological examination suggested undifferentiated NB with an atypical immunophenotype: PHOX-2B negative; MycN not amplified; S100 positive; ALK positive; c-Myc negative. Within the PREME project, WES was performed on the tumor and normal tissue. While this identified six possible candidate genes (Table 1), we selected the mutation c.A484G in the PSMC2 gene, as it can be targeted by Bortezomib, which had already been tested in the treatment of NB [18]. Restaging of the disease was performed through imaging techniques, biochemistry, bilateral bone marrow aspirates and trephines. Whole body (WB) MRI and 123I-MIBG scintigraphy showed widespread involvement of the soft tissues of the left arm; multiple skeletal metastases (orbital region, spine [particularly L4, L5 and sacrum], sternum, both humeri, pelvis, both femurs and tibias) and axillary lymph node metastases; trephine and myeloaspirate were negative for morphological infiltration but showed GD2-negative cellular infiltration on immunophenotypic study; the urinary acid vanilmandelic (VAM) value was in the normal range for the age.

A personalized second-line treatment regimen was proposed with the aim of achieving a second long period of complete remission and minimizing toxicity. The patient therefore received two courses of ICE chemotherapy, followed by significant clinical improvement and the disappearance of pathological uptake at all sites of disease on ^123^I-MIBG scintigrapy. A WB MRI showed a marked improvement of all skeletal and lymph node lesions. However, a pathological signal was still detectable in the left arm; trephine and myeloaspirates were negative. In September 2020, the patient underwent a third course of ICE, which was followed by grade 2 renal tubular toxicity according to the Common Terminology Criteria for Adverse Events v4.0 (CTCAE) and grade 3 hematological toxicity.

Disease evaluation performed after the three courses of chemotherapy showed a complete response of all known lesions, with a residual signal alteration in the upper arm.

Off-label target therapy with two courses of Bortezomib and Irinotecan and two courses of TEMIRI (Temozolomide, Irinotecan) was administered^18^ on the basis of the molecular alteration (*PSMC2* gene) found in the tumor tissue collected on relapse within the PREME project.

In January 2021, disease assessment was performed by means of ^123^I-MIBG scintigraphy, MRI and bone marrow examination, and showed complete remission (CR) (Figure 2).

Another course of Bortezomib plus Irinotecan was administered before consolidation therapy, which consisted of Thiotepa associated to Cyclophosphamide. Radiotherapy was performed at the site of relapse (30Gy). Treatment was completed with a total of five courses of anti-GD2. In October 2021, the patient achieved CR, which persisted over the five courses of Immunotherapy with Dinutuximab beta, which ended in March 2022; he is still in CR.

## 3. Discussion

In this report, we present two cases of patients evaluated in the Italian PeRsonalizEd MEdicine (PREME) cancer program for neuroblastoma (NB).

PREME was first instituted as a research project, and was subsequently approved as a clinical protocol. To date, 40 patients have been enrolled; these two cases were chosen because case 2 can be regarded as a positive example of the importance of promptly initiating targeted therapy during disease progression, while case 1, besides being the first case enrolled in PREME, represents a negative example, in that targeted therapy was introduced late, after a further relapse of the disease.

With regard to targeted therapy, the expert multidisciplinary working group of the MTB has defined targetable alterations as those associated with potential clinical benefit resulting from targeted therapy, whether alone or in combination with other therapies. This definition took into account both ongoing and completed trials and published data. In choosing the targeted therapy to administer, the possibility of receiving the drug for compassionate or off-label use was also discussed.

As in other programs of sequencing in pediatric oncology [12,19,20], we found that the overall mutational load was relatively low in comparison with adult cancers [21]. However, as the sequencing of NB patients has revealed a higher rate of mutations on relapse, this approach could constitute an opportunity for this subgroup of patients, who are without standardized treatment and have a poor prognosis [10,22,23,24,25]. Recently published analyses of genetic alterations across pediatric cancers have revealed many differences in the genomic alterations of pediatric versus adult cancers, underscoring the need for novel, mechanism-of-action-driven drugs specific for pediatric cancers [5,6,22,26]. In pediatric oncology, as in adults, molecular profiling programs aim to identify genetic alterations that are potentially actionable. In the two cases described here, this precision-medicine approach identified two different actionable mutations and allowed us to adopt a personalized therapeutic strategy with the possibility of inducing further complete remission.

The targeted sequencing of clinically relevant gene panels (about 100 genes) is widely used in the clinical setting to identify mutated genes that can be targeted by molecular drugs. However, the application of WES, as used in our two patients, presents advantages over targeted sequencing. First, the spectrum of clinically actionable genes is broader; indeed, it is now known that less frequently mutated genes can be good candidates for personalized treatment in cancer [27] Therefore, the targeted sequencing of specific genes is likely to be insufficient. Second, the completeness of WES also enables longitudinal investigations to be carried out, thereby facilitating the possible initiation of new clinical trials for previously unidentified cancer genes. On the other hand, the application of WES assumes that the lab group is equipped with sequencing platforms that rapidly generate a high output of sequences at low cost. Moreover, it assumes the availability of a team of bioinformaticians capable of developing analytic workflows that can fairly reliably prioritize potential clinically actionable mutations among a large number of clinically non-significant variants.

Despite the increased number of promising targets and drugs identified in pre-clinical studies, only a few anti-cancer agents have been evaluated in phase III clinical trials. For NB patients, the only non-immunological target agents used in front-line therapy are the ALK inhibitor (crizotinib) and the radiopharmaceutical iodine-labelled-meta-iodobenzylguanidine (NCT03126916). In addition, access to new anti-neoplastic agents for pediatric patients needs to be improved [3]. Indeed, many of the new drugs have not yet been approved for administration to children, not all patients can afford to pay for the off-label use of these high-cost agents, and compassionate use is rarely granted in the case of pediatric patients. The NB New Drug Development Strategy (NDDS) was launched with the aim of accelerating the development of new drugs for NB patients by prioritizing targets and drugs that should be introduced into pediatric clinical trials [28]. A major challenge for clinical trials in precision medicine (i.e., the AcSé-ESMART trial-European Proof-of-Concept Therapeutic Stratification Trial of Molecular anomalies in relapsed or refractory tumors) [29] is that they try to detect targetable mutations on the basis of the molecular profile in a cohort, regardless of histology. However, several studies have shown the limited efficacy of the target drugs in monotherapy; this is due both to the acquisition of secondary mutations, which lowers the binding affinity of the inhibitors, and to the pathway evasion strategies of cancer cells. In this regard, preclinical studies in NB have shown the synergistic activity of biological agents with backbone chemotherapy [30]. Since precision medicine first emerged as an approach to cancer treatment, biomarker-driven targeted therapies have become a reality. Umbrella and basket protocols allow the application of precision medicine within a tumor-agnostic approach. Today, there is interest in combination therapy with checkpoint inhibitors (NCT03837899, NCT03130959, NCT02304458) and in combinations involving chemotherapy, radiation or targeted radiation therapies (NCT02927769, NCT03445858, NCT02914405), which could reverse the “cold” immune environment. These types of trials increase the chance of achieving benefit for individual patients by providing evidence of efficacy earlier than trials that enroll unselected cohorts of patients. However, in many pediatric cancers, there is no clearly defined genomic profile.

Here, in both cases analyzed, WES detected somatic mutations that were potentially actionable. In case 1, the *CUL4A* somatic mutation was evidenced, while in case 2 the *PSMC2* proteosomal genomic alteration was detected.

*CUL4A* has been found to be over-expressed in many cancers and has been implicated in carcinogenesis. Moreover, epigenetic alterations have been functionally linked to the occurrence and development of ovarian cancer. The CXXC zinc finger protein 1 (CFP1) is an epigenetic regulator involved in DNA methylation and histone modification in mammalian cells. CFP1 promotes ovarian cancer cell proliferation and apoptosis, and the expression of CFP1 is reported to be affected by the CRL4 ubiquitin ligase complex [31]. Furthermore, cell cycle progression in mammals is modulated by two ubiquitin ligase complexes, CRL4 and SCF, which facilitate the degradation of chromatin substrates involved in the regulation of DNA replication. Indeed, selective interactions between replication origins and specific CRL components execute the DNA replication program and maintain genomic stability by preventing the re-initiation of DNA replication [32]. Finally, as *CUL4A* over-expression is common in pediatric medulloblastoma [13], it may constitute a promising target. CNS relapse is found in about 6.2% of all NB metastatic relapses [33]; it is a severe complication and is associated with a poor prognosis. Greater attention is currently being focused on the impact of new strategies (e.g., high-dose chemotherapy regimens and immunotherapy) on the pattern of relapse, especially as anti-GD2 antibodies do not penetrate the blood-brain barrier. Thus, the precise analysis of CNS recurrence is of major interest, as it may affect the design of future HR-NB strategies. One hypothesis is that skull lesions may directly involve the CNS; other possibilities are active penetration from the meninges, dissemination through cerebrospinal fluid, and hematogenous involvement. In our first case, the ^123^I-MIBG scintigraphy performed before maintenance therapy showed pathological uptake in only one skull lesion [33,34,35,36,37]. Advances in technologies, the development of agnostic approaches and the identification of targetable somatic tumor aberrations through genomic profiling programs enabled us to propose a personalized approach. The mutation found in the first patient was a typical CNS mutation that can be targeted by thalidomide/lenalidomide. So far, however, no somatic mutations of CUL4A have been reported in NB, but an enrichment of somatic variants has recently been observed in urothelial carcinoma with squamous differentiation (UCS) in comparison with urothelial carcinoma [38]. Another study has reported that the genomic amplification of CUL4A is associated with poor outcome [39]. These data suggest that genomic alterations of CUL4A can drive tumor initiation and progression; thus, CUL4A can be considered a good candidate therapeutic target, since drugs that inhibit its action are available.

It has been suggested that PSMC2 is a tumor promotor that induces HCC development and progression by directly interacting with ITGA6 [40]. Moreover, PSMC2 plays a critical role in OSCC progression by affecting pro-apoptotic protein expression and apoptosis pathways [41]. In addition, PSMC2 is reported to enhance RPS15A levels by targeting hsa-let-7c-3p, and then to activate the mTOR pathway, thereby promoting the progression of gastric cancer [42]. These findings indicate that targeting PSMC2 might be a promising strategy for cancer treatment.

The proteosomal genomic alteration detected in our second patient (*PSMC2* gene) is targetable by bortezomib, a proteasome inhibitor used in clinical practice, in association with melphalan and prednisone, for the treatment of patients affected by multiple myeloma [43].

It is to be underlined that several studies have reported unfavorable results in NB patients treated with single targeted therapy. The synergistic therapeutic benefit of combining target agents and chemotherapy has therefore been investigated in preclinical trials [44,45,46]. For instance, in a phase I clinical trial, lenalidomide was administered in combination with temozolomide in relapsed/refractory CNS patients and was well tolerated [15]. In another phase I clinical trial, bortezomib was administered in combination with irinotecan in patients with relapsed/refractory high-risk NB [18]. The combination was well tolerated, displaying a favorable toxicity profile and modest, but promising, clinical effect [18].

In summary, this article presents two examples of personalized therapeutic approaches which demonstrate that precision medicine strategies are possible and feasible even in patients with relapsed NB. Notably, complete remission (CR) was obtained in the second patient, in whom genomic profiling was performed at the time of relapse, while in the first patient, the personalized approach was considered only after the third-line therapy. This seems to highlight the importance of the timing of genomic profiling.

## 4. Materials and Methods

### 4.1. Patients and MTB Organization

Patients were identified in the PREME project as having relapsed NB without standard of care therapy. Participants provided written consent to the clinical project after the risks and benefits had been explained to them and/or their caregivers, including the potential disclosure of medically actionable secondary findings, defined as germline disease-causing mutations unrelated to the condition for which sequencing was being performed. Patients could opt in or out of the following: being informed of secondary findings and/or of the results of the present study; having their samples and/or data stored for future research, either with or without identifiers, and future contact.

The methods were performed in accordance with the relevant guidelines and regulations, and were approved by the CER (Regional Committee) first under the protocol ANTECER_Neuroblastoma: 15/12/2016, amendment 065_16/09/2019, and subsequently under the protocol EudraCT: 2022-000558-27.

### 4.2. Preparation and Characterization of Neuroblastoma Samples 

Biological samples (peripheral blood and tumor tissues) were centralized at the Laboratory of Experimental Therapies in Oncology, IRCCS Istituto Giannina Gaslini. Peripheral blood was collected in EDTA and used fresh for flow cytometry immunophenotyping and DNA extraction; tumor tissues were kept in culture medium (RPMI-1640) or stored in a freezing solution containing 90% serum and 10% DMSO until use (histological immunophenotyping and DNA extraction).

DNA was extracted from the tissue and blood samples by means of the QIAgen QIAamp DNA mini-kit in accordance with the manufacturer’s instructions (QIAgen, Hilden, Germany). The kit is designed to purify total DNA through the use of columns containing a silica-based membrane.

Briefly, the tumor tissues were first mechanically homogenized by means of the Tissue Lyser system (QIAgen, Hilden, Germany) and lysed by means of the kit lysis buffers and proteinase K. The solution obtained was loaded onto the kit columns, RNA was digested by adding RNase A, and finally the purified DNA was eluted with nuclease-free water. The DNA concentration was quantified through fluorometric assay using the Qubit platform (Thermo Fisher Scientific, Waltham, MA, USA), and DNA integrity was assessed by performing gel electrophoresis (0.8% agarose). Samples were stored at −20 °C until use. A genomic sequencing analysis was carried out by means of WES. Libraries were generated by using an Agilent Sure Select Human Whole Exon Kit v.6. Paired-end sequencing (2 × 150 bp) was performed by means of the Illumina HiSeq 1500 platform. Sequencing data [Tumor tissue and control data (reads)] were aligned against the human reference genome (GRCh37/hg19) by means of the Burrows-Wheeler Aligner (BWA) program. The alignment files were sorted by genomic coordinates by means of the Sam tools program; the same program was used to remove duplicate reads (aligned with identical genomic coordinates and resulting from DNA PCR amplification steps). Tumor tissue somatic variants [single nucleotide variants, SNVs, and small (up to 50 bp) insertions/deletions, INDELs] were identified by means of the GATK Mutect2 program. Finally, the lists of variants identified were annotated by means of ANNOVAR. The DNA from tumors and blood was sequenced to depths of 100× and 50×, respectively.

### 4.3. Data Interpretation and Reporting

Clinical WES was interpreted by the MTB and, approximately 30 days after testing, a report was generated. After the quality control of DNA sequences, only variants with at least 5% of tumor allele frequency were evaluated and studied.

For the purpose of clinical discussion, the report included variants with a pathogenicity score of at least 20 (calculated by CADD) and a CancerVar score greater than 0.80 in genes that potentially interact with anti-neoplastic drugs according to the DGIdb database (https://www.dgidb.org, accessed on 1 January 2019). The Therapeutic Target databases (dB) and The Cancer Molecular-Targeted Therapy dB were also examined in order to identify “potential target genes”. The MTB considered the genes to be targetable on the basis of literature evidence and the availability of experimental drugs, of age-appropriate information on dosing based on phase I studies, and of pre-clinical evidence of efficient targeting and/or potential clinical benefit.

## 5. Conclusions

As knowledge of tumor biology advances and sequencing becomes more feasible, it is mandatory to implement genome profiling both on diagnosis and on relapse in order to allow personalized treatment for patients affected by NB.

Identifying somatic alterations that are potentially actionable by new drugs may enable us to undertake patient-tailored therapy aimed at both improving survival and reducing systemic toxicity in relapsed/refractory NB patients.

As precision medicine programs are likely to be quite varied, perhaps one of the most interesting challenges will be to standardize the definition of what are/are not actionable targets. This aspect will be central to the success or otherwise of precision medicine.

## Figures and Tables

**Figure 1 ijms-23-11236-f001:**
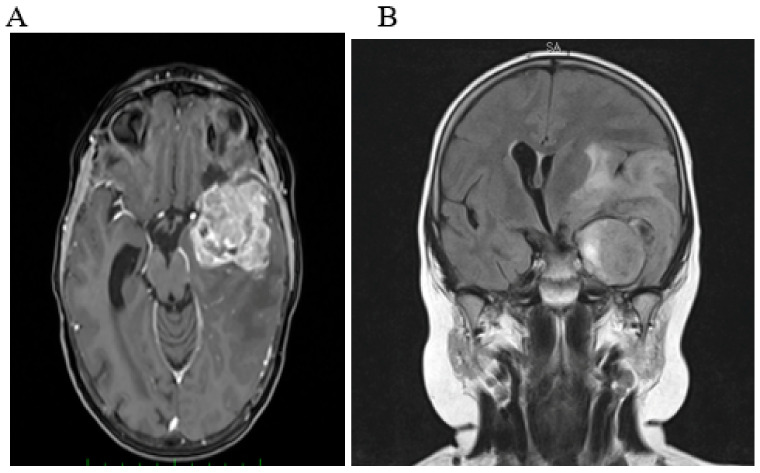
MRI axial 3D T1 gadolinium (Axial view of 3D T1-weighted gadolinium-enhanced MRI) (**A**) and coronal FLAIR (**B**): left temporal lobe intraparenchymal metastatic solid NB lesion, with inhomogeneous contrast enhancement, edema, mass effect, and midline shift.

**Figure 2 ijms-23-11236-f002:**
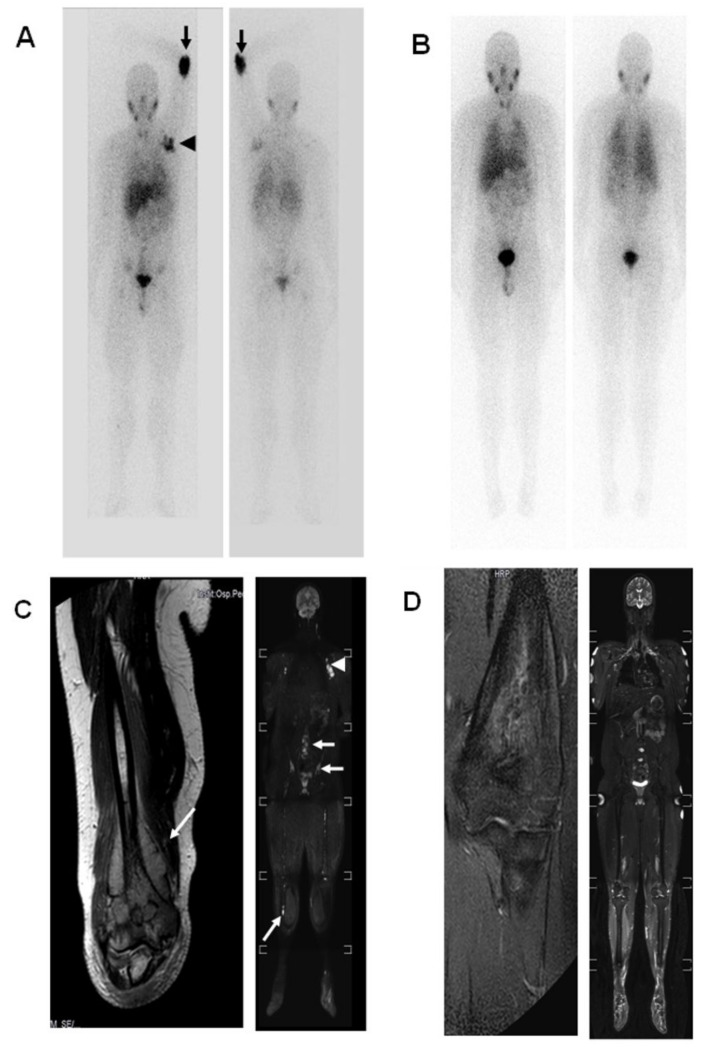
(**A**): Whole Body 123I-mIBG, anteroposterior and posteroanterior images. On relapse, scintigraphy revealed an intense MIBG uptake in the left elbow (arrow) and in the axillary hollow (arrowhead). A mild uptake was seen in the proximal femurs, iliac bones and lumbar spine. (**B**). Whole-body 123I-mIBG, anteroposterior and posteroanterior images. At the end of therapy, no MIBG alterations were present. (**C**). Magnetic Resonance Imaging (MRI). Coronal T2-Turbo SpinEcho (TSE) sequence of the elbow and whole-body coronal multiplanar reconstruction diffusion-weighted imaging with background body signal suppression (DWIBS) image. On relapse, MRI of the elbow showed a large lesion involving the distal humerus and soft tissue (arrow); Whole-body DWIBS revealed increased lymph nodes in the left axillary hollow (arrowhead) and several skeletal signal alterations in the lumbar vertebrae, iliac bones, femurs and right tibia (arrows). (**D**). MRI, coronal T2-weighted short time inversion-recovery (T2W-STIR) of the elbow and whole-body coronal T2-weighted short time inversion-recovery (T2W-STIR) image. At the end of therapy, elbow bone remodeling with periosteal reaction was seen; a mild bone marrow signal alteration was present, owing to red bone marrow conversion. At the end of treatment, a Whole-body STIR image showed no skeletal alterations. The MRI was performed on a 1.5 Tesla scanner (Achieva D-Stream, Philips).

**Table 1 ijms-23-11236-t001:** Candidate actionable genes.

Chr	Position	REF	ALT	Gene	Mutation Characteristics	COSMIC	SIFT	CADD	CancerVar	Drug	Interaction Types
**Case 1**											NA
chr2	240,139,318	G	C	OTOS	exon4, c.C122G, p.P41R		D	25	0.8762	CISPLATIN	NA
chr6	136,651,004	G	T	MAP3K5	exon11, c.C1768A, p.H590N		T	24.6	0.9925	HYDROXYUREA	NA
chr13	113,219,000	G	C	CUL4A	exon3, c.G320C, p.R107P		D	28.5	0.9925	LENALIDOMIDE, THALIDOMIDE, POMALIDOMIDE	inhibitor
chr16	49,789,532	C	T	ZNF423	exon2, c.G31A, p.A11T		D	23.2	0.9925	TAMOXIFEN	NA
**Case 2**											
chr2	141,143,477	C	A	LRP1B	exon67, c.G10516T, p.D3506Y		D	34	0.702	DOXORUBICIN	NA
chr3	134,920,343	C	A	EPHB1	exon12, c.C2158A, p.Q720K		D	32	0.842	VANDETANIB	inhibitor
chr4	72,623,854	C	A	GC	exon7, c.G736T, p.A246S		D	29.5	0.996	CETUXIMAB, CETUXIMAB	NA
chr7	93,065,322	G	C	CALCR	exon11, c.C1091G, p.S364C		D	25.8	0.9632	PRAMLINTIDE	agonist
chr7	103,003,194	A	G	PSMC2	exon6, c.A484G, p.T162A	ID=COSM484567	D	24.6	0.9959	IXAZOMIB CITRATE, CARFILZOMIB, BORTEZOMIB	inhibitor
chr20	43,703,716	A	G	STK4	exon11, c.A1363G, p.M455V		D	25.1	0.2034	BOSUTINIB	inhibitor

Chr: chromosome; REF: reference allele; ALT: altered allele; NA: not available.

## Data Availability

The raw sequence data generated during the current study are not publicly available, as no patient consent was required in order to deposit these sequencing data in a public repository. However, the data are available from the corresponding author on reasonable request.

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
