# Peer review of "Italian Precision Medicine in Pediatric Oncology: Moving beyond Actionable Alterations"

_ijms, 2022, doi:10.3390/ijms231911236_

Round 1
Reviewer 1 Report
Thanks for asking me to review this submission.
Precision medicine is an emerging and important area in paediatric oncology. In this context the submission is timely however I do have signficant concerns about the submission.
First there is too much clinical detail on the cases at the expense of the detail on the precision medicine program and how it works, how the decisions of the molecular tumour borad were made and applied to choosing the targeted therapy.
Second I think there are a number of major caveats about the identification and selection of "actionable targets". This is an important area. The target- drug database used, has been predominantly generated from adult cancer trials without a substantial number of paediatric cancer trials included. This likely limits the applicability of how the database can be used.
There are gaps in reasoning between the identification of a target on the WES & the choice to target that specific nucelotide change. A central difficulty also lies in the lack of orthoganal validation of each of the actionable targets - ie going back to demonstrate increased expression of CUL4A / PSMC2. More data about each of the nucleotide changes needs to be included in the table. For example - there is no information on which exon the nucleotide lies in, whether the expected amino acid change is conservative or non conservative, whether there is a predicted change in the structure of the protein, whether any of these changes have been identified as pathogenic in COSMIC etc. There is no orthogonal validation of the targeted mutation and the expected effect - eg in case 1 it is assumed in the case report that the CUL4A nucleotide variation results in overexpression & there is broad literature based justification for the choice of the specific therapy. There is no discussion around the known role of cullin in normal biology nor the effect of overepxression or knockdown. Apart from the linkage in the CCMT dB there appears to be no literature to link mutations in CUL4A and thalidomide / pomalidomide, there is no consideration of CUL4A SNV as a passenger mutation acquired during the course of resistant disease versus a driving mutation, nor any disucssion whether this or a similar CUL4A mutation has been described in relapsed / refractory neuroblastoma or any other relapsed malignancy. In case 2, the assumption is also drawn that because the gene encodes a component of the S26 proteasomal complex that a proteasome inhibitor is the appropriate choice. However there is no clear data presented that proteasomal inhibitors bind the protien subunit encoded by this gene (PSMC2) and a broad assumption is drawn that because the gene is in the pathway it's therefore an actionable target.
It is not possible to conclude in either case what effect (if any) the selected therapeutic agent had on the patient outcome as both were delivered in the context of other therapy.
There is no real discussion around the pros and cons of dealing with the multiple hits in the WES data and how these were considered and balanced in the molecular tumour board. More importantly though there's no real discussion around what are "actionable targets" in WES / WGS data. Which SNVs are clinically significant, possible drivers as opposed to passenger mutations that arise in or are selected for in refractory / relapsed cancer? This is the central issue in personalised medicine - the accurate and timely identification of clear actionable targets whilst at the same time dismissal of the noise of multiple SNVs in genomic data.
I remain unconvinced about the real quality of the identified actionable targets. I would suggest that in a newly diagnosed high risk patient the identification of either of these SNVs would not result in a change in treatment. I accept that a different choice has been made in the relapse setting but the level of supporting data for therapeutic efficacy should remain as high in the relapse setting as any other to reduce the risk of potentially ineffective, possibly toxic and probably expensive therapy. For example, the identification of an ALK mutation/amplification is currently regarded as a high quality actionable target based on supporting laboratory & clinical data & would be actionable at both diagnosis and relapse. There is a enough data to support taregting ALK with appropriate inhibitors. Yet there is minimal supporting literature for either target (CUL4A / PSMC2) in this submission and it's not clear that SNVs in these genes have been previously identifed in relapsed / refractory neuroblastoma or any other paediatric cancer. It's also not clear that there any publsihed cases / lietrature linking these alterations with a direct reponse to the agents in either labortaory or clinical reports outside of the CCMT database. So the overall supporting evidence for each actionable SNV & drug is sparse. Would the MTB regard either / both these SNVs as being taregetable if they were identified in other patients or were they the best available for these specific patients & would they be disregarded in other contexts for other patients?
There's no real discussion around the benefits and drawbacks of the molecular strategy and the lack of orthoganal validation of targets. Would there be more robust data if there were additonal data available eg RNA expression, methylation, protein expression etc? Should other sources of data & databases be used to identify targets (eg COSMIC) and should there be other published evidence to support the choice of an actionable SNV and the linked drug? There could be a mpre extensive discussion on the benefits and drawbacks of the strategy chosen to link nucleotide changes with drugs using the CCMT dB. Is any other precision medicine program using this approach & how are potential actionable targets in this database validated if there is limited or no supporting published litearture to back up the potential gene-drug interactions in the CCMT database? How did the molecular tumour prioritise the molecular findings and how were they presented to the treating clincian in the report? Was the choice to target specific genes made by the molecular tumour board, by the individual clinician after recieving a report or jointly together? Would either of the actionable targets identified here also be identified by other precision medicine teams as being actionable? I think this is an area of likely great variation between precision medicine programs and perhaps one of the most interesting aspects of the submission - standardisation of what are / aren't actionable targets will be central to the success or not of precision medicine.
Overall it's a potentially interesting case report but I think there are significant limitations as currently written. I think it would be more compelling submission if there was less focus on the cases and more focus on the intrinsic challenges of applying genomic data into clinical care with a closer focus on the decision making processes around actionability, the individual SNVs and their predicted impact, choice of one actionable SNV over another & how the molecular tumour board & clinicians balance these challenges.
I have made some comments on the draft of the submission.

Author Response
Referee: First there is too much clinical detail on the cases at the expense of the detail on the precision medicine program and how it works, how the decisions of the molecular tumour board were made and applied to choosing the targeted therapy.
AAs: thanks to reviewer’ comments, few redundant clinical details have been delated, while adding explanation about the choice of the targeted therapy.
Added to the Discussion section
In this report, we present 2 cases of patients evaluated in the Italian PeRsonalizEdMEdicine (PREME) cancer program for neuroblastoma (NB). PREME was born as a research project and was then approved as a clinical protocol. In total, 40 patients have been enrolled to date and these 2 cases have been chosen because case 2 can be considered a positive example on the importance of rapid introduction of targeted therapy during disease progression, while case 1, besides being the first case enrolled in PREME, it represents a negative example because the target therapy was introduced late after a further relapse of the disease.
Moreover, during the virtual meeting of the MTB, after presentation of clinical cases, the recommendations were discussed by all working group members to reach a final consensus. The expert multidisciplinary working group defined targetable alterations as those associated with potential clinical benefit with a targeted therapy alone or in combination with other therapies. In all cases, trials ongoing, completed, and data published were evaluated. For the final decision to choosing the targeted therapy the possibility to receive the drug as compassionate or off label use has been also discussed.
(See also below for further explanation).
Referee: Second I think there are a number of major caveats about the identification and selection of "actionable targets". This is an important area. The target- drug database used, has been predominantly generated from adult cancer trials without a substantial number of paediatric cancer trials included. This likely limits the applicability of how the database can be used.
AAs. Beyond the Therapeutic Target databases (dB), we used the database DGIdb (Nucleic Acids Res. 2021 Jan 8;49(D1):D1144-D1151) that provides different types of information, not only it includes active clinical trials but also many other evidences regarding the biology of gene in relation with the drug under investigation. DGIdb uses a combination of expert curation and text-mining and drug-gene interactions have been mined from DrugBank, PharmGKB, Chembl, Drug Target Commons, TTD, and others. Our plan was to use DGIdb to obtain a list of potential druggable genes to be discussed in the tumor board meeting and verify the possibility to apply an alternative treatment option.
Referee: More data about each of the nucleotide changes needs to be included in the table. For example - there is no information on which exon the nucleotide lies in, whether the expected amino acid change is conservative or non conservative, whether there is a predicted change in the structure of the protein, whether any of thesechanges have been identified as pathogenic in COSMIC etc.
AAs. In accordance, we have now added in the Table 1 the nucleotide changes and the CancerVar pathogenic score, recently published on Science Advance (PMID: 35544644), designed specifically to prioritize druggable mutations. Both mutations in CUL4A and PSMC2 showed and a high CancerVar index (0.99). We have also added COSMIC annotation data and the SIFT pathogenic score that predicts whether an amino acid substitution affects protein function. Among the ten selected variants only that in PSMC2 has been already observed in one tumor tissue, particularly in the cancer kidney. Finally, all variants, expect that in MAP3K5, were predicted to alter the protein function by SIFT.
Referee: there is no consideration of CUL4A SNV as a passenger mutation acquired during the course of resistant disease versus a driving mutation, nor any discussion whether this or a similar CUL4A mutation has been described in relapsed / refractory neuroblastoma or any other relapsed malignancy
AAs. So far, no somatic mutations of CUL4A have been reported in neuroblastoma but recently an enrichment of somatic variants have been observed in urothelial carcinoma with squamous differentiation (UCS) when compared with urothelial carcinoma (PMID: 35918248). Another study reports that the genomic amplification of CUL4A is associated with poor outcome (PMID: 26684807). These data suggest the genomic alterations of CUL4A can drive tumor initiation and progression. We have now commented this in Discussion section.
Added to the Discussion section
So far, no somatic mutations of CUL4A have been reported in neuroblastoma but recently an enrichment of somatic variants have been observed in urothelial carcinoma with squamous differentiation (UCS) when compared with urothelial carcinoma (PMID: 35918248). Another study reports that the genomic amplification of CUL4A is associated with poor outcome (PMID: 26684807). These data suggest the genomic alterations of CUL4A can drive tumor initiation and progression and thus it can be considered as good candidate therapeutic target since drugs that inhibit its action are available.
Referee: There is no real discussion around the pros and cons of dealing with the multiple hits in the WES data and how these were considered and balanced in the molecular tumour board. More importantly though there's no real discussion around what are "actionable targets" in WES / WGS data. Which SNVs are clinically significant, possible drivers as opposed to passenger mutations that arise in or are selected for in refractory / relapsed cancer? This is the central issue in personalised medicine - the accurate and timely identification of clear actionable targets whilst at the same time dismissal of the noise of multiple SNVs in genomic data.
AAs. Our medicine precision program is designed to include not only the most studied and known cancer driver genes and mutations. It is now known that less frequently mutated genes can be good candidates for the personalized treatment in cancer. We have thus opted to apply WES that for sure has the cons togenerate multiple variants to be analyzed but we employ computational methods that help usfor realizing an effective clinical analysis and interpretation of WES data. We have now discussed the pos and cos in the Discussion section.
Added to the Discussion section
The targeted sequencing of clinically relevant gene panels (about 100 genes) is largely used in clinical setting to identify mutated genes that can be targeted with molecular drugs. However, the application of WES presents advantages over targeted sequencing. First, the spectrum of clinically actionable genes is increasing; indeed, it is now known that less frequently mutated genes can be good candidates for the personalized treatment in cancer (PMID: 24836576) Therefore, the targeted sequencing of specific genes are likely to be insufficient. Second, the completeness of WES also allows for longitudinal queries on the possible opening of new clinical trials for previously unidentified cancer genes.On the other hand, the application of WES assumes that the lab group is equipped with sequencing platforms that generate a high output of sequences at low times and costs and with a team of bioinformaticians capable of developing analytic workflows for prioritizing, with a relative high reliability, potentially clinically actionable mutations among a large number of not clinically significant variants.
Referee: There's no real discussion around the benefits and drawbacks of the molecular strategy and the lack of orthogonal validation of targets.
AAs. As described in the section “Data interpretation and reporting”, our general strategy was to divide in three main steps: i) prioritization of genes with variants of clinical significance obtained by using dedicate pathogenic scores; ii) selection of genes that are candidate targets of antineoplastic drugs by using dedicate databases; iii) Molecular Tumor Bord meeting toevaluate the eventual treatment options based on biological evidences of the prioritized genes, literature evidences and availability of experimental drugs, of age-appropriate information on dosing based on phase I studies, and of pre-clinical evidence of efficient targeting and/or potential clinical benefit. This strategy allows MTB to discuss on large but at same time reliable set of candidate target genes.
Although, at the present time, we cannot offer orthogonal validation of targets, we are already generating preclinical models (such as Patient-derived xenograft (PDX) mouse models and 3D cellular models) to be used for the recruitment and treatment of the next patients.
AAs: Finally, in agreement with the Referee’ comment this sentence has been added in the Conclusion section:
This is an area of likely great variation between precision medicine programs and perhaps one of the most interesting aspects will be standardisation of what are/aren't actionable targets. This aspect will be central to the success or not of precision medicine.
Reviewer 2 Report
​- In the presentation of the case 1 the author could deeper describe the UL4A's role in modifying chromatin that is largely related to DNA repair, the same comment is valid for c.A484G in the PSMC2 in case 2
-The authors presented 2 cases of patients evaluated in the Italian PeRsonalizEd MEdicine (PREME) cancer program for neuroblastoma; could be interesting if they underline in the discuss why they selected these two and how many patients are involved in the PREME program.
Author Response
Referee: ​- In the presentation of the case 1 the author could deeper describe the UL4A's role in modifying chromatin that is largely related to DNA repair, the same comment is valid for c.A484G in the PSMC2 in case 2
AAs: in agreement with the referee’ suggestion a deeper description has been added in the Discussion section.
Added:
CUL4A has been found to be over expressed in multiple cancers and implicated in carcinogenesis. Moreover, thus can represent a promising target. epigenetic alterations have been functionally linked to ovarian cancer development and occurrence. The CXXC zinc finger protein 1 (CFP1) is an epigenetic regulator involved in DNA methylation and histone modification in mammalian cells. CFP1 promotes ovarian cancer cell proliferation and apoptosis, and the expression of CFP1 was affected by CRL4 ubiquitin ligase complex (Cancer Gene Ther 2022 Jul 21. doi: 10.1038/s41417-022-00503-z). Furthermore, cell cycle progression in mammals is modulated by two ubiquitin ligase complexes, CRL4 and SCF, which facilitate degradation of chromatin substrates involved in the regulation of DNA replication. Indeed, selective interactions of replication origins with specific CRL components execute the DNA replication program and maintain genomic stability by preventing re-initiation of DNA replication (Nat Commun 2018 Jul 17;9(1):2782. doi: 10.1038/s41467-018-05177-6). Finally, CUL4A over expression is common in pediatric medulloblastoma, thus it can represent a promising target.
…..
However, so far, no somatic mutations of CUL4A have been reported in NB but recently an enrichment of somatic variants have been observed in urothelial carcinoma with squamous differentiation (UCS) when compared with urothelial carcinoma (PMID: 35918248). Another study reports that the genomic amplification of CUL4A is associated with poor outcome (PMID: 26684807). These data suggest the genomic alterations of CUL4A can drive tumor initiation and progression and thus it can be considered as good candidate therapeutic target since drugs that inhibit its action are available.
It has been proposed that PSMC2 is pinpointed as a tumor promotor to interfere HCC development and progression via interacting with ITGA6 directly (Cell Death Discov 2021 Aug 19;7(1):217. doi: 10.1038/s41420-021-00585-y). PSMC2 plays a critical role in OSCC progression through affecting pro-apoptotic protein expression and apoptosis pathways (Cell Cycle 2022 Mar;21(5):477-488. doi: 10.1080/15384101.2021.2021722). Moreover, PSMC2 enhanced RPS15A levels by targeting hsa-let-7c-3p, and then activated mTOR pathway, thereby promoting the progression of gastric cancer (Oncogenesis 2022 Mar 7;11(1):12. doi: 10.1038/s41389-022-00386-7). These findings indicate that targeting PSMC2 might be a promising strategy for cancer treatment.
Referee: -The authors presented 2 cases of patients evaluated in the Italian PeRsonalizEdMEdicine (PREME) cancer program for neuroblastoma; could be interesting if they underline in the discuss why they selected these two and how many patients are involved in the PREME program.
AAs: we thank the reviewer for the suggestion.
Added to the Discussion:
In this report, we present 2 cases of patients evaluated in the Italian PeRsonalizEdMEdicine (PREME) cancer program for neuroblastoma (NB). PREME was born as a research project and was then approved as a clinical protocol. In total, 40 patients have been enrolled to date and these 2 cases have been chosen because case 2 can be considered a positive example on the importance of rapid introduction of targeted therapy during disease progression, while case 1, besides being the first case enrolled in PREME, it represents a negative example because the target therapy was introduced late after a further relapse of the disease.
Round 2
Reviewer 1 Report
Many thanks for asking me to review the re-submission.
I continue to have ongoing major reservations about this submission. Although the authors have responded to the prior review I do not feel that they have taken the opportunity to step back and critically consider the presentation and interpretation of the data. I do not feel that the submission is more nuanced and overall I think the conclusion isn't justified by the data.
I remain unconvinced that either of the cases demonstrate significant evidence of identifying and targeting the proposed mutations identified by WES and which resulted in unequivocal clinical benefit to the patient. Neither case as presented supports the conclusion "The two patients were then treated according to these actionable alterations with promising results." Both patients received multiagent therapy and it's clearly possible that the same / similar clinical responses might have been achieved without the therapy recommended by the molecular tumour board. I was hoping that the authors would have revised their conclusion in the re-submission.
The revised manuscript doesn't adequately grapple with the choice of a targetable mutation and how the molecular tumour board really weighed up whether the individual molecular results were drivers / passengers, clinically relevant / not clinically relevant, whether there was clear significance of the selected mutations in relapsed neuroblastoma or whether they should have been regarded as targetable at all. It appears that the recommendation of therapy was as much driven by the availability of agents that had been used in paediatric oncology practice & could be combined with chemotherapy rather than a strong molecularly based rationale. This is the interpretation drawn from the statement: "The expert multidisciplinary working group defined targetable alterations as those associated with potential clinical benefit with a targeted therapy alone or in combination with other therapies". It appears that the emphasis of the molecular tumour board is to recommend a treatment. I would ask the authors to consider under what circumstances the tumour board would reach the conclusion that there are no targetable findings to recommend.
Overall I think there has been a failure to adequately address the issue raised previously "Which SNVs are clinically significant, possible drivers as opposed to passenger mutations that arise in or are selected for in refractory / relapsed cancer? This is the central issue in personalised medicine - the accurate and timely identification of clear actionable targets whilst at the same time dismissal of the noise of multiple SNVs in genomic data."
The additional detail for both CUL4a & PSMC2 emphasise roles in other cancers but not neuroblastoma which ultimately raises questions as to whether either pathway was important & targetable.
Ultimately the other issue that hasn't really been addressed is the suitability of the databases & platforms used for treatment recommendations. The databases appear to be predominantly derived from adult cancer practice & although the authors provided some more technical detail around the databases & resources used, they haven't addressed the issue of whether adult dataset should be used or is applicable in the paediatric cancer setting.
Author Response
Reviewer: I remain unconvinced that either of the cases demonstrate significant evidence of identifying and targeting the proposed mutations identified by WES and which resulted in unequivocal clinical benefit to the patient. Neither case as presented supports the conclusion "The two patients were then treated according to these actionable alterations with promising results." Both patients received multiagent therapy and it's clearly possible that the same / similar clinical responses might have been achieved without the therapy recommended by the molecular tumour board.
AAs: We agree with the Referee that the sentence "The two patients were then treated according to these actionable alterations with promising results" is too strong and we accept her/his suggestion to revise our conclusion adding the final concern: “As a note of caution, because both patients received multiagent therapy it is possible that similar clinical responses might have been achieved without the therapy recommended by the molecular tumour board.” Moreover, “with promising results” has been deleted.
Reviewer: It appears that the recommendation of therapy was as much driven by the availability of agents that had been used in paediatric oncology practice & could be combined with chemotherapy rather than a strong molecularly based rationale. This is the interpretation drawn from the statement: "The expert multidisciplinary working group defined targetable alterations as those associated with potential clinical benefit with a targeted therapy alone or in combination with other therapies".
AAs: We partially agree with the Referee because, as modified already after the first revision, this case report presents the results of two patients out of 40 and for 18 of them a recommended therapy based by a molecular rationale was found. Indeed, the tumour board reached the conclusion that for the other cases there are no targetable findings to recommend although the availability of various anti-cancer drugs. However, to better clarify this point, the above statement "The expert multidisciplinary working group...” has been modified in the Discussion section, accordingly.
Reviewer: Overall I think there has been a failure to adequately address the issue raised previously "Which SNVs are clinically significant, possible drivers as opposed to passenger mutations that arise in or are selected for in refractory / relapsed cancer? This is the central issue in personalised medicine - the accurate and timely identification of clear actionable targets whilst at the same time dismissal of the noise of multiple SNVs in genomic data."
AAs: We apologize for not adequately addressing this important point during the previous revision. The following sentences have been added in the MS. To distinguish possible cancer drivers from passenger SNVs, computational approaches that aim to predict driver mutations according to their frequency of occurrence in a cohort of samples, or according to their predicted functional impact on protein sequence or structure, can be used (Genome Med. 2014 Oct 14;6(10):81. doi: 10.1186/s13073-014-0081-7. eCollection 2014. PMID: 25360158). Here, since we are not analyzing a cohort of patients but only single cases enrolled at different times, we employed diverse algorithms that predict the impact of the genetic alterations on protein sequence and structure. We would emphasise that we also used a score (Science Advances, Sci Rep. 2021 Dec 7;11(1):23551. doi: 10.1038/s41598-021-02671-8. PMID: 34876593) specifically designed for the prioritization of SNVs that can drive malignant transformation.
Reviewer: The additional detail for both CUL4a & PSMC2 emphasise roles in other cancers but not neuroblastoma which ultimately raises questions as to whether either pathway was important & targetable.
AAs: These details were added because requested by the 2nd Reviewer.
Reviewer: … they haven't addressed the issue of whether adult dataset should be used or is applicable in the paediatric cancer setting.
AAs: As already requested in the first round of revision, in the revised MS, we added other databases&platforms, in our opinion, suitable for treatment recommendation in the pediatric cancer setting. However, in the pediatric setting (including neuroblastoma), mutations are rare and target drugs even rarer. Moreover, the molecular targets for pediatric cancers validated with clinical or pre-clinical studies are very few. All the international pediatric precision medicine platforms, for the selection of molecular drugs based on genetic alterations, are mostly based on clinical and preclinical studies (in vivo and in vitro) performed on other pediatric and, mostly, non-pediatric cancers.